# Accuracy of the Low-Dose ACTH Stimulation Test for Adrenal Insufficiency Diagnosis: A Re-Assessment of the Cut-Off Value

**DOI:** 10.3390/jcm8060806

**Published:** 2019-06-05

**Authors:** Laura Maria Mongioì, Rosita Angela Condorelli, Federica Barbagallo, Rossella Cannarella, Sandro La Vignera, Aldo Eugenio Calogero

**Affiliations:** Department of Clinical and Experimental Medicine, University of Catania, Via S. Sofia 78, 95123 Catania, Italy; lauramongioi@hotmail.it (L.M.M.); federica.barbagallo11@gmail.com (F.B.); roxcannarella@gmail.com (R.C.); sandrolavignera@unict.it (S.L.V.); acaloger@unict.it (A.E.C.)

**Keywords:** adrenal insufficiency, cortisol levels, low-dose ACTH test stimulation

## Abstract

Background: The clinical practice shows that many low-dose ACTH-stimulation tests have a false positive result. The aim of the study was to determine the diagnostic accuracy of a low-dose ACTH-stimulation test in the diagnosis of adrenal insufficiency and to define its optimal cut-off. Methods: We analyzed data from 103 patients undergoing 1 µg ACTH-stimulation test. Four patients had adrenal insufficiency (AI) upon follow up: Two primary, and two secondary AI. Cortisol serum levels were evaluated at time 0, 20’, and 30’ after the injection of 1 µg i.v. of ACTH. The sensitivity, specificity, accuracy, and positive and negative predictive values of the test were calculated for both 20’ and 30’ sampling. The receiver operating characteristic (ROC) curve was obtained to assess the sensitivity and specificity of low-dose ACTH-stimulation test in the diagnosis of adrenal insufficiency at different cut-off values. Results: Considering 500 nmol/L as the standard cut-off value, low-dose ACTH stimulation test showed a 100% sensitivity and a 67.3% specificity, with a high rate of false positive results. ROC curve analysis showed that the cut-off of 401.5 nmol/L is the best compromise between sensitivity (100%) and specificity (93.9%). Conclusions: By using a cut-off value of 401.5 nmol/L for the low-dose ACTH stimulation test, the number of false positive patients decreased significantly, but the sensitivity remained high.

## 1. Introduction

Adrenal insufficiency (AI) is a rare, life-threatening disease caused by either primary adrenal failure [1] or by hypothalamic–pituitary impairment of the corticotropic axis alone or in combination with other anteropituitary hormone deficiencies (CPHD) [2]. AI, even to a mild degree, is associated with increased mortality; therefore, a timely diagnosis is essential [3].

The diagnosis of AI is often difficult due to an onset of non-specific symptoms over the period of time (sometimes many years). In past years, the most validated test for the integrity of the hypothalamic–pituitary–adrenal (HPA) axis was the cortisol response to insulin-induced hypoglicaemia, known as the insulin stress test (IST). However, this test is largely considered too dangerous to perform. According to the Endocrine Society, the corticotropin (ACTH) stimulation test is the new diagnostic “gold standard” for the diagnosis of AI, although there is still some debate about the cut-off value of cortisol after stimulation. Moreover, different protocols are used in clinical practice with respect to the duration of the test, intramuscular or intravenous drug administration, and dose of corticotropin administered [4]. Usually, the standard short ACTH test measures cortisol levels at 30 and 60 minutes after the administration of 250 µg of corticotropin as an intravenous bolus. Traditionally, peak cortisol levels below 500 nmol/L (18 µg/dL) are suggestive of AI, but the results are assay-dependent [4]. In clinical practice, low-dose (1 µg of corticotrophin) ACTH stimulation testing is often used, since it gives results comparable or even more sensitive than that of high-dose (250 µg) testing [4,5,6].

The aim of this study was to evaluate if a cut-off different from 500 nmol/L for 1 µg ACTH stimulation test could increase its sensitivity and specificity in the diagnosis of AI.

## 2. Material and Methods

The test was performed in 103 consecutive patients (20 males and 83 females, mean age: 39.6 ± 1.47 years) attending the Division of Andrology and Endocrinology, Teaching Hospital “Policlinico-Vittorio Emanuele,” University of Catania between July 2016 and July 2018, with signs and/or symptoms suggestive of AI and cortisol levels between 82.75 nmol/L and 500 nmol/L. As the symptoms are similar in both primary and central forms, it is not always possible to distinguish between the two forms. Although ACTH levels are useful for this purpose, in some cases they do not allow this objective to be achieved. In this study, before testing, 49% (*n* = 50) of patients were suspected to have a primary form of AI, since their basal cortisol levels were <276 nmol/L, and ACTH levels were normal or high, and many of them had chronic autoimmune thyroiditis and/or other autoimmune diseases. Instead, 17% (*n* = 18) of patients were suspected to have a central form of AI, as they presented hypothalamic-pituitary diseases (mainly pituitary adenoma) currently or in the past years. Finally, for 33% (*n* = 34) of patients it was not possible to hypothesize the etiology of the possible AI (cortisol levels <276 nmol/L and ACTH normal levels in absence of other suggestive data).

Each patient who participated in this study was asked to sign a written informed consent form and the Intra-Divisional Ethic Committee of the Endocrinology Section approved the research protocol.

We excluded from this study patients who had used glucocorticoids in the last two weeks before the test was performed. The presence of severe comorbidities was also considered the exclusion criterion.

Cortisol serum levels were evaluated at zero, 20, and 30 minutes after the intravenous injection of 1 µg of 1-24 tetracosactid (1 µg ACTH stimulation testing). Every test was performed at 8.00 AM. Serum cortisol concentrations were determined by an enzyme-linked immunocolorimetric assay in an automated procedure.

Sensitivity, specificity, positive and negative predictive values, and accuracy of the test were calculated. The receiver operating characteristic (ROC) curve was obtained to assess the sensitivity and specificity of 1 µg ACTH stimulation testing in the diagnosis of AI at different cut-off points.

### 2.1. Statistical Analysis

Sensitivity, specificity, positive (PPV) and negative (NPV) predictive values, and accuracy were calculated using the number of patients’ true-positive (TP), true-negative (TN), false-positive (FP) and false-negative (FN) results. PPV (calculated as TP / (TP + FP) was defined as the likelihood that a subject with a positive test (peak cortisol below the specific cut-off point) was affected with primary AI. NPV (calculated as TN / (TN + FN) was defined as the likelihood that a subject with a negative test (peak cortisol above the specific cut-off point) has a normal adrenal function. By repeated-measures analysis of variance (ANOVA) with a Greenhouse–Geisser correction, we compared the serum cortisol levels of each patient at 0, 20, and 30 minutes. The receiver operating characteristic (ROC) curve was obtained to assess the sensitivity and specificity of the 1 µg ACTH stimulation test in the diagnosis of AI at different cut-off values for 20- and 30-minute serum cortisol levels during the test. The area under the ROC curve (AUC) was also calculated. The ROC curve at 30 minutes was used to determine the optimal cut-off value. McNemar’s test was used to compare the number of FPs between the standard cut-off of 500 nmol/L and the optimal cut-off value determined through the ROC curve.

A statistical analysis of the data was performed using the Statistical Package for the Social Sciences version (SPSS) 23.0 for Windows software program (IBM Corp., Armonk, NY, USA). Results with a *p* value of less than 0.05 were considered to be statistically significant.

## 3. Results

One female patient dropped out the study, since she did not initially report that she took glucocorticoids the day before the test. Table 1 summarizes the principal features of the whole population studied and of patients with AI.

Of the remaining 102 patients, 66 patients showed a normal response to ACTH stimulation, with peak cortisol levels greater than 500 nmol/L (18 µg/dL). Among these, seven patients showed a peak cortisol level of greater than 500 nmol/L at 20 minutes and a value lower than 500 nmol/L at 30 minutes. We considered them as responders (Group A). The remaining 36 patients showed a peak cortisol level below 500 nmol/L; hence, they were considered to be non-responders (Group B). After the execution of ACTH stimulating test, every patient was clinically followed. After the execution of ACTH stimulating test, every patient was clinically followed. Four patients in the Group B were diagnosed for AI (true positive): a female patient with hypopituitarism caused by previous pituitary macroadenoma that had been surgically-removed (secondary AI, baseline cortisol 196 nnmol/L, ACTH 5.3 pg/mL), a male patient with thalassemia, hypogonadism and GH deficiency (secondary AI, baseline cortisol 199 nnmol/L, ACTH 28.8 pg/mL), and two female patients with presumed autoimmune adrenalitis (baseline cortisol 28 nmol/L and 135 nmol/L and ACTH were 6.4 pg/mL and 65 pg/mL). The last two patients also underwent high-dose (250 µg) ACTH test stimulation, and cortisol levels did not reach the cut-off value of 500 nmol/L.

Figure 1 shows serum cortisol values at 0, 20, and 30 minutes after 1 µg ACTH administration. A repeated-measures ANOVA with a Greenhouse–Geisser correction showed that cortisol levels were significantly different at 0, 20, and 30 minutes (*F*(1520, 155.089) = 440.41, *p* < 0.01). Post-hoc tests using the Bonferroni correction revealed that cortisol levels at zero minutes were statistically significantly different versus cortisol levels after 20 and 30 minutes of 1 µg ACTH administration (*p* < 0.01). The test also showed a statistically significant difference between cortisol levels at 20 and 30 minutes (*p* < 0.01). These results showed the importance of the evaluation of cortisol levels at 20 minutes from ACTH administration.

Considering 500 nmol/L as a cut-off value, low-dose ACTH stimulation showed a 100% sensitivity and a 67.3% specificity, enabling the identification of all patients with AI, but with a high rate of FP results. It correctly identified AI only in 11.1% of patients with positive tests (PPV) and normal adrenal function in 100% of patients with negative tests (NPV). The test had an accuracy of 68.6%.

Figure 2 shows the ROC curve at 20 and 30 minutes. ROC curve analysis revealed the cut-off of 401.5 nmol/L to be the best compromise between sensitivity (100%) and specificity (93.9%). In this way, the sensitivity remained the same, but the specificity significantly increased (i.e., from 67.3% to 93.9%). McNemar’s test showed that, by lowering the cut-off value from 500 nmol/L to 401.5 nmol/L, the number of FPs significantly decreased (*p* < 0.01). In addition, the accuracy of the test significantly increased (i.e., from 68.6% to 94.1%).

Table 2 summarizes performances of 1 µg ACTH stimulation for cut-off levels of 500 nmol/L and 401.5 nmol/L.

## 4. Discussion

A short ACTH stimulation test is actually the “gold standard” for the diagnosis of AI. According to the Endocrine Society guidelines, cortisol levels should be measured after an intravenous administration of ACTH at the standard dose of 250 µg for adults and children over two years of age, a dose of 15 µg/kg for newborns, and a dose of 125 µg for children under two years, respectively [4], but different protocols are used in clinical practice. Some debates concern the fact that the dosage of 250 µg of ACTH is supra-physiological and is capable of generating FN responses in the case of mild forms of AI [3]. Therefore, it has been suggested that a more suitable adrenal evaluation can be obtained, by using a lower dose of ACTH (1 µg), and by evaluating cortisol levels at 20 to 30 minutes after ACTH injection [5].

In the present study, cortisol serum levels were evaluated at zero, 20, and 30 minutes after the intravenous injection of 1 µg of 1-24 tetracosactid. We found that cortisol levels at zero minutes were significantly different from those measured at 20 and 30 minutes after the administration of 1 µg of ACTH. The test also showed also a statistically significant difference between cortisol levels at 20, and 30 minutes, respectively. These results showed the importance of the evaluation of cortisol levels 20 minutes after ACTH administration, since some patients showed peak cortisol levels >500 nmol/l at this time-point and <500 nmol/l after 30 minutes. Thus, the evaluation only at 30 minutes could be misleading (increasing the positive rate).

To avoid exposure to supraphysiological doses, we chose 1 µg ACTH stimulation testing, since studies in the literature have shown its reproducibility [5] and comparable results with the high-dose test [4]. In 2015, Ospina et al. published a systematic review and meta-analysis about the diagnostic accuracy of the high (250 µg) and 1 µg (i.e., low-dose) ACTH stimulation test in the diagnosis of AI, both central and primary. Both tests showed similar diagnostic accuracy (low sensitivity and high specificity) in the diagnosis of central AI [6]. Regarding primary forms, the data available for the 1 µg test were insufficient to draw conclusions. However, the use of 1 µg testing could be useful for the early diagnosis of the initial or partial stages of primary AI [7].

Regarding the timing of blood sampling for cortisol evaluation, it has been reported that, the lower the dose of ACTH is, the earlier the timing of the peak cortisol levels is [8] and, for the 1 µg ACTH stimulation test, the peak cortisol levels occurred after about 20 minutes [9].

Previous studies have shown that cortisol values of 497 nmol/L and above, in healthy subjects undergoing 1 μg ACTH stimulation, are appropriate, and many authors have previously used this cut-off value [10,11,12]. We considered 500 nmol/L as a cut-off value, which was used to distinguish responders from non-responders. We found that 1 µg ACTH stimulation testing presented a 100% sensitivity, a 67.3% specificity, and a 68.6% accuracy, allowing us to identify patients with AI, but with a high rate of FP results. We also found that, by reducing the cut-off to 401.5 nmol/L, the test sensitivity did not change, whereas the specificity significantly increased and the number of FPs significantly decreased. Moreover, after the ACTH stimulation test, every patient was clinically followed and only four patients (3.9%) were diagnosed for AI (true positive). This finding is in agreement with epidemiological data, which indicates a prevalence of about 100 to 140 cases per million and an incidence of 4:1,000,000 per year in Western societies [4].

To our knowledge, this is the first study that evaluated sensitivity and specificity of 1 µg ACTH stimulation testing by using a lower cut-off.

Although peak cortisol levels below 500 nmol/L (18 µg/dL) are usually suggestive of AI, the normal cortisol response to ACTH testing remains not completely defined, possibly due to the various subject- and test-condition-related factors, such as the use of oral contraception, the degree of fat mass, and the methods of assay [13]. Some authors evaluated the response to ACTH stimulation testing using higher cut-off values and found several subnormal responses, resulting in further increased FP responses [5]. Tordjman et al. performed ROC analysis to evaluate the following different potential cut-off values after 1-µg ACTH stimulation: 275, 415, 470, 500, 550, 605, and 690 nmol/L. The authors found that, by using a stimulated cortisol level of above 500 nmol/L, the test had 94.7% sensitivity, 90% specificity, and 91% accuracy [14]. We showed that, by using 401.5 nmol/L as a cut-off value, test sensitivity was 100%, test specificity was 93.9%, and test accuracy was 94.1%. Hence, this value of cortisol improved all parameters in comparison with the other studies.

## 5. Conclusions

Low-dose (1 µg) ACTH testing is useful for AI diagnosis, by avoiding exposure on the part of the patient to a supraphysiological ACTH dosage. Although cortisol values of less than 500 nmol/L are usually considered suggestive of AI, the correct cortisol cut-off to use remains under debate. We showed that, by using a cut-off value of 401.5 nmol/L for low-dose ACTH stimulation, the number of FPs could be significantly diminished, maintaining an elevated sensitivity and thus improving the accuracy. Further studies with a greater number of patients are useful to confirm these findings.

### Limitation

The study lacks a comparison with the 250 µg ACTH stimulation test. Although there are many debates on this diagnostic test, today it has to be considered as the gold standard for the diagnosis of AI. Further studies will be needed.

## Figures and Tables

**Figure 1 jcm-08-00806-f001:**
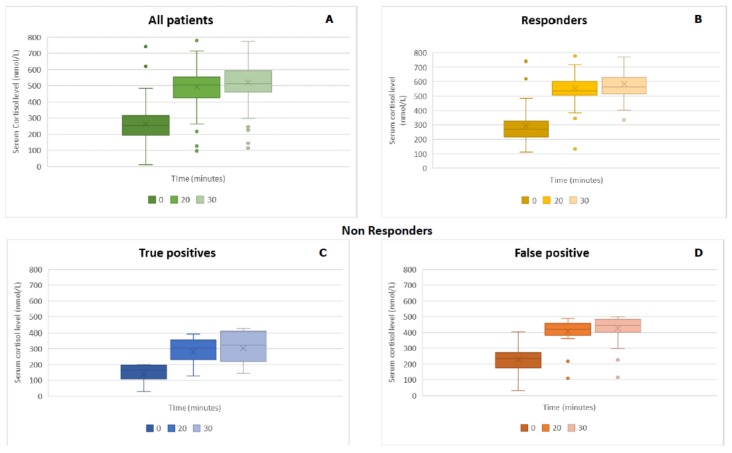
Serum cortisol values at 0, 20, and 30 minutes after corticotropin (ACTH) administration in (**A**) all patients; (**B**) the responder group (cortisol peak >500 nmol/L); (**C**) true positive (cortisol peak <500 nmol/L, AI diagnosis); and (**D**) false positive (cortisol peak <500 nmol/L, but no AI diagnosis).

**Figure 2 jcm-08-00806-f002:**
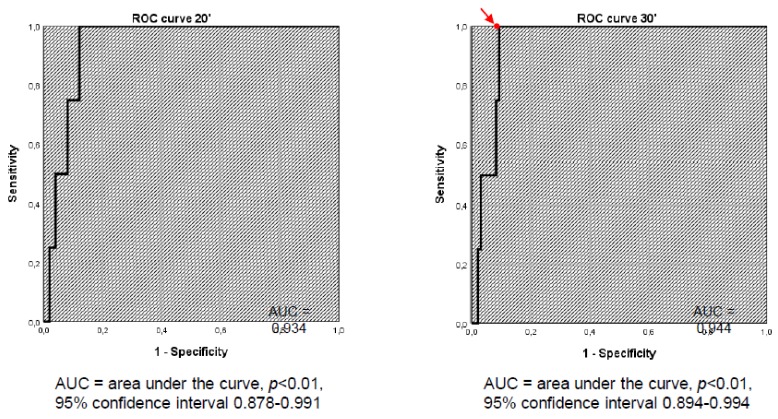
Receiver operating characteristic (ROC) curve of serum cortisol levels at 20 and 30 minutes after ACTH stimulation.

**Table 1 jcm-08-00806-t001:** Demographic characteristics of the cohort of patients included in the study and in those with AI.

Parameters	All Patients	AI
Female	82	3
Male	21	1
Age	39.6 ± 14.5	40.3 ± 14.2
Weight	68.8 ± 15.5	78.4 ± 7.4
BMI	26.7 ± 5.5	30.9 ± 3

AI = Adrenal insufficiency.

**Table 2 jcm-08-00806-t002:** Performances of 1 µg ACTH stimulation for cut-off levels of 500 nmol/L and 401.5 nmol/L.

Parameters	1 µg ACTH Stimulation Test	
	Cut-off of 500 nmol/L	Cut-off of 401.5 nmol/L
False negative	0	0
False positive	32	3
Sensitivity (%)	100	100
Specifity (%)	67.3	93.9
Accuracy (%)	68.6	94.1
PPV (%)	11.1	40
NPV (%)	100	100

PPV = Positive Predictive Value; NPV = Negative Predictive Value.

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
