# Peer review of "Accuracy of the Low-Dose ACTH Stimulation Test for Adrenal Insufficiency Diagnosis: A Re-Assessment of the Cut-Off Value"

_jcm, 2019, doi:10.3390/jcm8060806_

Reviewer 1 Report

"If the authors used AI cases determined by ACTH stimulation test, it is obvious that they got high sensitivity and 100% specificity of this test on AI cases. I don't think it is an appropriate approach to evaluate the clinical effectiveness of ACTH test on the diagnosis in which this test is already used."

Author Response

In the clinical practice, the ACTH stimulation test is used when the presence of AI is suspected (see line 83). The aim of present study was not to evaluate the usefulness of this test, since it is considered the “gold standard” to diagnose AI by the Endocrine Society (see line 68). This study was undertaken to identify a new cut-off value that would give greater sensitivity and specificity to the test. To accomplish this, we tested patients with symptoms suggestive of AI and then followed them clinically over time. Among the patients who showed cortisol levels below 500 nmol/l after ACTH stimulation test, only 4 had AI and they were treated appropriately (true positive) (see lines 131-137). By using a cut-off value of 401.5 nmol/l, the test sensitivity, specificity and accuracy increased so this could improve the clinical approach and the management of patients with suspected AI.

Reviewer 2 Report

The authors analyzed their experience with a low dose 1mg ACTH stimulation test for adrenal insufficiency. They suggest that a level below 401.5 nmol/L would be the best for defining adrenal insufficiency.

 Methods

The authors need to define either pre-test or post- test criteria for adrenal insufficiency (AI) as primary or secondary .  The results of the tests cannot be used to define a true positive.

 Results

The demographics of the patients studied should be given in a table ( age, sex, weight, bmi ).  The demographics of the patients with adrenal insufficiency and the diagnostic reasons for the AI given in the table.

The results of the cortisol levels with SEM at each time point should be given in a figure comparing true negatives from true positives (AI). The current figure 1 may be modified to compare AI from comparators which would have more relevance.

The authors should give a pre-test reason to define responders from non- responders as they do not use this classification for any further clinical evaluation. Since the classification arbitrarily separates those with lower levels from those with higher levels, statistical tests are irrelevant.

Also statistical test comparing baseline to 20 and 30 minute cortisol levels are to be expected in a stimulation test and need not be reported.

The major issue is how to define sensitivity and specificity with only 4 true positives .

Further analysis and descriptions may clarify their conclusions.     

Author Response

Methods:

         We evaluated the cortisol response to ACTH stimulation in patients with symptoms suggestive of AI. As the symptoms are similar in both primary and central forms, in the clinical practice it is not always possible to distinguish between the two forms. Although ACTH levels are useful for this purpose, in some cases they do not allow to achieve this objective. In this study, before testing, 49% (n=50) of patients were suspected to have a primary form of AI, since their basal cortisol levels were<276 nmol/L and ACTH levels were normal or high and many of them had chronic autoimmune thyroiditis and/or other autoimmune diseases. Instead, 17%(n=18) of patients were suspected to have a central form of AI, as they presented hypothalamic-pituitary diseases (mainly pituitary adenoma) currently or in the past years. Finally, for 33% (n=34) of patients it was not possible to hypothesize the etiology of the possible AI (cortisol levels<276 nmol/L and ACTH normal levels in absence of other suggestive data). This has been added in the revised version of the manuscript (please see lines 84-92).

After ACTH stimulation test, every patient was clinically followed and only four patients in the group of non-responders resulted to have AI (true positive): a female patient with hypopituitarism due to previous pituitary macroadenoma surgically-removed, a male patient with thalassemia, hypogonadism and GH deficiency and two female patients with chronic autoimmune thyroiditis and presence of adrenal cortex autoantibodies (ACA). The last two patients also underwent to high-dose (250 µg) ACTH test stimulation and cortisol levels did not reach the cut-off value of 500 nmol/l. This was added in lines 131-137.

 Results: 

·         We added the table requested (please see table 1). The diagnosis of AI are reported in the text (please see lines 131-137)

·         Thank you for your comments that gave us the opportunity to improve Figure 1 to show the results more clearly. In particular, it is evident that cortisol values after ACTH stimulation tests are lower in true positive patients compared to false positive. We preferred to use the representation with box plots rather than mean ±SEM, because we believe it is clearer and easier to understand, but if the editor feels differently, we will modify the figure accordingly.

·         About a pre-test reason to define responders from non-responders, the literature has established that a peak cortisol levels below 500 nmol/L (18 µg/dL) after ACTH stimulation test has to be considered suggestive of AI. This was already written in lines 73-74. Thus, from the clinical point of view, a cortisol value above 500 nmol/L excludes the diagnosis of AI, whereas a peak<500 nmol/l is suggestive for the presence of this disease.

·         We compared the cortisol values in response to the test to have a more complete statistical evaluation. Moreover, to our knowledge no study in literature has analyzed the differences between 20 and 30 minute cortisol levels after ACTH stimulation test. The results of our analysis highlighted the importance of the evaluation of cortisol levels 20 minutes after ACTH administration, since some patient showed peak cortisol levels >500 nmol/l at this time-point and<500 nmol/l after 30 minutes. Thus, the evaluation only at 30 minutes could be misleading (increasing the positive rate). This has been added in lines 143-144 and 171-174.

·         Regarding the definition of test sensitivity and specificity with only 4 true positives, it must be considered that AI is a rare disease with a prevalence of about 100 to 140 cases per million and an incidence of 4:1,000,000 per year in Western societies (Bornstein et al., J Clin Endocrinol Metab. 2016; 101, 364-89). Similar prevalence and incidence have been reported by other studies (Laureti et al., J Clin Endocrinol Metab. 1999; 84, 1762;  Betterle et al., Endocr Rev 2002; 23, 327–364; Løvas et al., Clin Endocrinol. 2002; 56,787–791). In consideration of the epidemiology of AI, we think that four true positive patients, that is 3.9% of our sample size, should not be considered an insufficient number. These considerations have been added in the Discussion (please see lines 192-195).

Reviewer 3 Report

In this study the authors determine the diagnostic accuracy of the low-dose ACTH stimulation test in the diagnosis of adrenal insufficiency.  They have analysed data form 103 patients undergoing 1 ug ACTH stimulation test and by using the receiver operating characteristic (ROC) curve conclude that the cut-off value for cortisol at 401.5 nmol/L is the best compromise between sensitivity (100%) and specificity (93.9%) and therefore decreases the number of false positive patients compared to the cut-off value of 500 nmol/L.   The whole manuscript is very well written and has a great practical value.  The statistical methods are appropriate and well described.  The graphs are very clear.  In my opinion this manuscript is ready for publication.

 Minor comment:

In line 134 remove one “also”

Author Response

Thank you for your comments.

The mistake was corrected.

Round  2

Reviewer 1 Report

I had already asked as following; "If the authors used AI cases determined by ACTH stimulation test, it is obvious that they got high sensitivity and 100% specificity of this test on AI cases. I don't think it is an appropriate approach to evaluate the clinical effectiveness of ACTH test on the diagnosis in which this test is already used." On the other hand, the authors replied: This study was undertaken to identify a new cut-off value that would give greater sensitivity and specificity to the test. To accomplish this, we tested patients with symptoms suggestive of AI and then followed them clinically over time.

Thus, you subjected the patients with symptoms suggestive of AI, while did not examine the responsiveness of cortisol to ACTH among normal control, latent adrenal insufficiency pure Addison`s disease and panhypopituitarism. 

Author Response

The aim the study was to evaluate if a cut-off different from 500 nmol/L could increase test sensitivity and specificity (this has been made clearer in the revised version of the manuscript, please see lines 78-79). Moreover, ACTH test stimulation is a second level diagnostic procedure, useful when symptoms of AI are present and when cortisol values range between 82.75 nmol/L and 500 nmol/l (see line 84). Outside of this range, low dose ACTH testing is not indicated. Therefore, this test was performed only in patients with cortisol value >82.75 nmol/L but<500 nmol/l. The results of the test using the standardized and well codified cut-off of 500 nmol/L showed that 36 out 103 patients (35%) had an abnormal result (i.e. they should receive the diagnosis of AI). All these 36 patients with a presumptive diagnosis of AI were followed for not less than 12 months and 32 (89%) of them did not have AI. Only four out of the 36 patients with low responsiveness to ACTH had AI. The novelty of this study is that another cut-off value (401.5 nmol/L) decreased drastically the number of false positive patients from 32 to 6, thus increasing the specificity of the test from 77.3% to 93.9% without decreasing the sensitivity rate (please see lines 147-156). On the basis of these considerations, we did not examine a control group.

Reviewer 2 Report

The authors followed up on modification of their  manuscript.

There were only 4 subjects with AI and so most of the analysis derives from an analysis of a normal range of 1mcg cortrosyn stimulation test is greater than 401.5 nmol/L at 30mintues. 

Abstract. Line 14 add comment: Four patients had AI upon follow up, two (or 3-as per line 103 below ) primary, and one (or 2) secondary AI.

Line103  Add and correct this statement. Four patients  were diagnosed with AI, one secondary (baseline cortisol ---,ACTH ---), two with presumed autoimmune adrenalitis (baseline cortisols ---,and ACTH were---- and ---) and one with thalassemia was primary (or secondary (baseline cortisol---and  ACTH was---) .

Table 1. Spelling: Specificity

Figure 2. What is red dot with arrow?  If this is the cut point, then point should be 1-specifity (1-93.9%)

Line116. I am unclear about this. If levels were over 401.5 at 20mintues but below 401.5 at 30mintues were they considered positive or negative?  How many times out of the 4 with AI did this combination occur?

Line 154 This seems irrelevant if only cut off 401.5 is chosen.  This line may be deleted.

Author Response

·         Thank you for your comments. We added the comment in the abstract and we modified the statement at line 103. We specified the disease diagnoses of the four AI patients as it was required in the major revision phase.

·         Table 1: There was a mistake. We added the correct table 1. We corrected the spelling mistake in table 2.

·         Figure 2: we corrected the mistake.

·         Lines 116 and 154: We considered positive at the test only patients with cortisol levels<500 nmol/L at 20 and/or 30 minutes from ACTH administration. For this reason, we  preferred to maintain the sentence at line 154. Subsequently, we performed ROC analysis to assess the sensitivity and specificity of the 1-µg ACTH stimulation test in the diagnosis of AI at different cutoff values for 20- and 30-minute serum cortisol levels during the test. We found that 401.5 nmol/L was the best cut-off for sensitivity and specificity. At the end of the analysis we found that among the all the four patients with AI had cortisol levels<401.5 nmol/L at 20 and 30 minutes.